# Ex Vivo Fluorescence Confocal Microscopy for Intraoperative Examinations of Lung Tumors as Alternative to Frozen Sections—A Proof-of-Concept Study

**DOI:** 10.3390/cancers16122221

**Published:** 2024-06-14

**Authors:** Max Kamm, Felix Hildebrandt, Barbara Titze, Anna Janina Höink, Hagen Vorwerk, Karl-Dietrich Sievert, Jan Groetzner, Ulf Titze

**Affiliations:** 1Department of Pathology, Medical School and University Medical Center OWL, Klinikum Lippe, Lung Cancer Center Lippe, Bielefeld University, 32756 Detmold, Germany; max.kamm@klinikum-lippe.de (M.K.); felix.hildebrandt@klinikum-lippe.de (F.H.); barbara.titze@klinikum-lippe.de (B.T.); 2Department of Diagnostic and Interventional Radiology, Medical School and University Medical Center OWL, Klinikum Lippe, Lung Cancer Center Lippe, Bielefeld University, 32756 Detmold, Germany; anna.hoeink@klinikum-lippe.de; 3Department of Pneumology, Respiratory and Sleep Medicine, Klinikum Lippe Lemgo, Lung Cancer Center Lippe, 32657 Lemgo, Germany; hagen.vorwerk@klinikum-lippe.de; 4Department of Urology, Medical School and University Medical Center OWL, Klinikum Lippe, Bielefeld University, 32756 Detmold, Germany; karl-dietrich.sievert@klinikum-lippe.de; 5Department of Thoracic Surgery, Klinikum Lippe Lemgo, Lung Cancer Center Lippe, 32657 Lemgo, Germany; jan.groetzner@klinikum-lippe.de

**Keywords:** confocal microscopy, lung cancer, frozen sections, digital pathology

## Abstract

**Simple Summary:**

Intraoperative consultation is frequently used in clinical practice to establish the initial diagnosis of lung cancer when preoperative histological confirmation fails. The ability to distinguish benign and malignant lesions established frozen sections as the gold standard to guide surgical procedures. Ex vivo fluorescence confocal microscopy is a novel fully digital microimaging technique that enables timely examinations of fresh tissue without loss. The method proved to be an effective and safe method for diagnosing and subtyping lung cancer in our study and is therefore a promising alternative to frozen sections. It preserved the tissue as native material for subsequent examinations, which is an advantage in the diagnosis of small tumors and for biobanking. The acquired digital image data in cross-platform formats formed a suitable basis for telepathological procedures, e.g., second opinions by external specialists.

**Abstract:**

Background: Intraoperative frozen sections (FS) are frequently used to establish the diagnosis of lung cancer when preoperative examinations are not conclusive. The downside of FS is its resource-intensive nature and the risk of tissue depletion when small lesions are assessed. Ex vivo fluorescence confocal microscopy (FCM) is a novel microimaging method for loss-free examinations of native materials. We tested its suitability for the intraoperative diagnosis of lung tumors. Methods: Samples from 59 lung resection specimens containing 45 carcinomas were examined in the FCM. The diagnostic performance in the evaluation of malignancy and histological typing of lung tumors was evaluated in comparison with FS and the final diagnosis. Results: A total of 44/45 (98%) carcinomas were correctly identified as malignant in the FCM. A total of 33/44 (75%) carcinomas were correctly subtyped, which was comparable with the results of FS and conventional histology. Our tests documented the excellent visualization of cytological features of normal tissues and tumors. Compared to FS, FCM was technically less demanding and less personnel intensive. Conclusions: The ex vivo FCM is a fast, effective, and safe method for diagnosing and subtyping lung cancer and is, therefore, a promising alternative to FS. The method preserves the tissue without loss for subsequent examinations, which is an advantage in the diagnosis of small tumors and for biobanking.

## 1. Introduction

Lung cancer is the most frequently diagnosed cancer and the leading cause of cancer death worldwide, responsible for almost 2.5 million new cases and an estimated 1.8 million deaths in 2022 [1]. The WHO classification of lung tumors is based on histology, supported by immunohistochemistry and molecular techniques [2]. Due to differences in metastatic pattern, recurrence, and survival between the histological subtypes, accurate assessment of pulmonary tumors is essential for the selection of optimal oncological treatment and surgical procedures. The diagnosis is based on clinical examination, imaging procedures (X-ray, Computed tomography), and various invasive methods, mostly fiber-optic bronchoscopy combined with endobronchial lymph node sonography (EBUS), with the aim of cytological or histological confirmation of malignancy [3].

The preoperative diagnosis of pulmonary lesions through transbronchial or transthoracic biopsy can be difficult in cases with small tumor sizes or due to the location of tumors. Meta-analyses showed a diagnostic yield of 75% for bronchoscopic approaches and 93% for percutaneous procedures (9 Studies, 7345 patients) for small lesions < 2 cm [4] and a pooled sensitivity of 0.67 for Bronchial Brushing Cytology (17 studies, 2538 patients) [5]. Thus, intraoperative examinations are frequently used in patients with inconclusive preoperative diagnoses in order to establish an initial diagnosis for the adequate surgical strategy. Intraoperative histology is insufficient in subtyping lung cancers and distinguishing primary lung cancers from metastases [6] as time-consuming immunohistology is needed in many cases. Nevertheless, it is an important diagnostic approach for guiding surgical procedures due to its overall ability to distinguish benign from malignant lesions.

Hematoxylin&Eosin (H&E)-stained intraoperative frozen sections have traditionally been used to classify pulmonary lesions, to determine the adequacy of surgical resection margins, and to evaluate whether the tissue obtained contains diagnosable tumor material [7] for companion immunohistochemistry and molecular pathology. Frozen sections of pulmonary lesions pose high demands on the pathologists involved because inflammatory atypia and histological artifacts can simulate malignancies [8]. Furthermore, considerable technical limitations such as severe distortion of the tissue architecture, ice-crystal formations, and collapse of the alveolar spaces during cryosection lead to a 2–13.1% false-negative rate and a 0–0.2% false-positive rate as compared with diagnoses made with post-FFPE tissue material [9]. It should be noted that frozen section is a technically demanding and personnel-intensive method that is not available in every facility.

Several digital microimaging techniques have been developed in recent years for examinations of native tissues as alternatives to frozen sections [10]. Laser scanning fluorescence confocal microscopy (FCM) is a highly advanced technique that is already commercially available (VivaScope, München, Germany and Diagnostic Inc., New York, NY, USA). The system enables the acquisition of high-resolution microscopic images with and without fluorescent dyes (ex vivo and in vivo FCM). The technique is established in routine dermatopathology [11,12] in Moh’s surgery of skin tumors for intraoperative control of surgical margins [13] and in the context of inflammatory dermatoses [14,15]. Its suitability for extracutaneous surgical pathology was recognized comparatively late [16], and only in recent years have the instruments been established in a growing number of centers [17]. Most experience has been gained in the field of urological practice [18], particularly in the biopsy diagnosis of prostate carcinomas [19,20,21] and as an alternative to frozen sections in prostatectomies [22,23,24]. The work of our research group documents that the technique can be integrated into routine pathology methods [25] and has proven to be a valuable tool for biopsy diagnostics of other organ systems, e.g., from the liver [26,27] and for biobanking of prostate carcinomas [28].

To our knowledge, the present work is the first systematic feasibility study on the suitability of FCM for the intraoperative classification of lung tumors. The assessment of surgical margins was the subject of a separate project. The primary endpoint of this study was the determination of the biological nature (benign vs. malignant) of lung tumors using FCM scans in comparison to paraffin-embedded material. The second endpoint was the suitability of the method for histological subtyping of pulmonary neoplasms and its limitations.

## 2. Materials and Methods

### 2.1. Participants and Available Surgical Specimens

A total of 57 patients aged between 36 and 83 years (mean 65 ± 10.7; 32 males and 25 females) who underwent thoracic surgery in the Lung Cancer Center Lippe (Department of Thoracic Surgery Lemgo, Lemgo, Germany) participated in this study. The indications for surgical intervention were suspicious masses in the lungs on preoperative imaging in 24 patients. In 21 patients, a non-small cell lung carcinoma was confirmed histologically or cytologically in the preoperative diagnosis. A further 14 patients with a history of carcinoma and new PET-positive round lung lesions underwent surgery with suspected metastases or primary lung tumors. Two patients had two separate lesions, so 59 surgical specimens (30 wedge resections, 5 segmental resections, and 24 lobectomies) were available for examination in the study. All participating patients were informed about the examinations in the study and provided signed informed consent. All investigations in the study were performed in accordance with the ethical principles of the WMA Declaration of Helsinki. The study was approved by the local ethics committee (Ref. 2020-029-f-S, Medical Association of Westphalia-Lippe, Münster, Germany).

### 2.2. Surgical Pathology Dissection and Histological Examinations of the Lung Specimens

The specimens were sent by express delivery service to the Department of Pathology at the Klinikum Lippe Detmold, Germany, for intraoperative examinations and companion histopathological workup. The macroscopic examinations were carried out in accordance with the requirements of current guidelines. In wedge and segmental resections, the staple lines were removed and the resulting parenchymal margin was marked with ink. The enclosed tumors were lamellated perpendicular to the surface and their size and distance from the resection margin were measured. In the case of lobectomies, staple lines were first removed, and the bronchial and vascular margins were shaved and processed in frozen sections. The airways were then opened, and the parenchyma was sectioned in a plane that best demonstrated the individual pathology including the tumors, their sizes, and their relations to the airways, the parenchyma, the pleura, and margins. The intraoperative frozen sections were examined by experienced pathologists (MK, FH, UT) within 30 min and the findings were transmitted by phone.

The resection specimens were then fixed in formalin for 24 h. The next day, macroscopic examination and dissection were performed by experienced pathologists (MK, FH, UT) from the Lung Cancer Center Lippe. The additionally taken samples were FFPE-processed together with the materials from the intraoperative examinations. The H&E-stained slides were available for the first report on the 2nd day after receipt of the material.

The carcinomas were routinely examined histochemically for mucin production (PAS/DPAS on Ventana Benchmark™ Special Stains platform) and immunohistologically for the expression of cytokeratin (CK) 7, p40 and TTF-1. Depending on the histomorphology, the panel was supplemented by neuroendocrine markers (CD56, synaptophysin, chromogranin A) and the proliferation marker Ki67. In case of metastases or other tumor entities, organ-specific markers (CDX-2, GATA-3, Pax-8, and others) were available. The results of the immunohistologies for the final subtyping of the tumors were usually available after a further 2–3 working days. All immunohistological stains were carried out on Ventana Benchmark™ (Ventana Medical Systems, Tucson, AZ, USA) platforms.

The indications for additional predictive immunohistological (PD-L1) or molecular examinations (driver mutations) were decided in interdisciplinary tumor boards and carried out at the Department of Pathology (Oncomine Focus Assay, Thermo Fisher, Waltham, MA, USA).

### 2.3. Ex Vivo Fluorescence Confocal Microscopy

The VivaScope 2500-G4 (VivaScope GmbH, Munich, Germany) microscope used in the study was a confocal laser scanning microscope designed for the intraoperative examination of unfixed tissues. The native materials require a short pretreatment with Acridine Orange (AO). The tissue is not damaged by the procedure and is available for all subsequent histological, immunohistological, and molecular studies. AO is commonly used as a fluorescent dye to stain live tissues for intraoperative confocal endomicroscopy and fluorescence microscopy that intercalates with nucleic acids [29]. The staining protocol in this study included three steps: 1. surface protein denaturation with ethanol for 10 s; 2. nuclear staining with AO solution (0.6 mM; Sigma-Aldrich^®^, St. Louis, MO, USA) for ~30 s; 3. removal of excess dye in saline solution for ~10 s.

The tissue was aligned flat on a special slide and placed in the automated stage of the instrument. The microscope was controlled exclusively by software. Illumination was provided by a long-wavelength laser (638 nm) for imaging cytoplasmic and extracellular structures (reflection mode) and a short-wavelength laser (488 nm) for fluorescence excitation of the AO-stained cell nuclei (fluorescence mode). A built-in software algorithm combined the information from reflection and fluorescence mode in real-time in a pseudo-colored image [30]. The resulting digital scans resembled hematoxylin-eosin-stained frozen sections, in which cytoplasmic/extracellular structures were depicted in red and cell nuclei in blue. The saturation of the staining could be modulated through software-controlled intensity adjustment of the excitation lasers.

The microscope was equipped with a 38× water immersion objective with a numerical aperture of 0.85, delivering magnifications of up to 550×. According to the manufacturer, specimens up to 2.5 × 2.5 cm in size could be examined. In addition, the VivaScope 2500-G4 is equipped with a digital camera that provides macroscopic images of the tissue samples prior to microscopic examination. The macroscopic images correlate exactly with the microscopic images and allow navigation and selection of the area to be examined.

### 2.4. Study Design

After the acquisition of the frozen sections, additional samples were taken from the tumors for FCM analysis to avoid delaying the routine intraoperative diagnosis. Digital images were recorded simultaneously during the intraoperative examinations (UT) and saved as anonymized image data.

All examinations for the study took place after the completion of routine examinations and had no impact on the clinical care of the patients (Figure 1). To evaluate the performance of FCM examinations, the diagnoses were compared with the results of frozen sections (FS) and conventional microscopy of H&E-stained slides of FFPE-processed material (H&E). The agreements of the individual methods with the final diagnosis (FD) were evaluated. The digital FCM scans were evaluated intraoperatively by a senior pathologist from the tumor center (UT). A second senior pathologist (BT) with a specialization in pulmonary pathology reevaluated the FCM scans in a blinded manner after all routine examinations had been completed. Additionally, the H&E-stained paraffin sections of the FCM materials were assessed in a similar manner by another experienced pathologist (MK) from the tumor center in a blinded fashion. The classification of the sections/scans was carried out according to a standardized schema with predefined diagnostic categories, which were later translated into nominal scales for subsequent statistical analysis. Initially, the nature of the lesions examined was grouped into two categories: 1—neoplastic; 2—reactive. In the second step, the nature of the neoplastic tumors was classified into three categories: 1—benign; 2—malignant; 3—uncertain. Additional free-text diagnoses were to be provided for the categories of benign tumors and uncertain lesions. Malignant tumors were further classified into four groups: 1—squamous cell carcinomas, 2—adenocarcinomas, 3—neuroendocrine tumors, and 4—other/special entities. The documented results of the frozen sections as well as the final diagnoses (immunohistological subtypes) were retrospectively transferred into the same matrix by another pathologist (FH).

The specimens were worked up in accordance with the recent guidelines (green pathway). Frozen sections (red pathway) and FCM examinations (blue pathway) were carried out intraoperatively in parallel on separate tumor samples. The results of the intraoperative examinations and those of the FFPE-processed residual material were compared with the final diagnoses.

As a measure of the practicality of FCM, the rate of inter-observer agreement between both examiners (UT vs. BT) for the assessments of tumor behavior (benign vs. malignant) and preliminary subtyping was determined. Furthermore, agreements with frozen sections and conventional H&E morphology were examined. Finally, the diagnostic performance of each method was evaluated in comparison to the final diagnosis and qualitatively analyzed for any relevant differences.

### 2.5. Statistical Analyses

The diagnoses of each method (FCM, FS, H&E, FD) were standardized as ratings on nominal scales. The evaluation was conducted using error tables, which allowed the analysis of relative diagnostic frequencies. For the detection of malignant tumors, the sensitivity, specificity, and negative and positive predictive values of the individual methods were determined and compared with each other.

The levels of agreement with the final diagnosis were measured using Cohen’s kappa [31]. The number of matching ratings (p0) was expressed as a quotient of the expected random number of matches (pe), which can assume values between 0 (random matches) and 1 (perfect match). The interpretation was based on the Landis and Koch categories (κ < 0: less than chance agreement; κ = 0.01–0.20: slight agreement, κ = 0.21–0.40: fair agreement, κ = 0.41–0.60: moderate agreement, κ = 0.61–0.80: substantial agreement, and κ = 0.81–1.0: almost perfect agreement) [32].

## 3. Results

### 3.1. Findings in the Final Histology

The final histological diagnoses based on conventional histology including special stains (PAS/DPAS) and immunohistochemistry are shown in Table 1.

The final diagnoses are shown after a complete pathological workup with representative sampling and supplementary immunohistology.

A total of 48/59 (81%) specimens revealed neoplastic lesions, of which 2 lesions were benign pulmonary hamartomas and 1 case showed a pleural solitary fibrous tumor of uncertain biological nature. A total of 45 specimens contained malignant tumors of which 27 were pulmonary adenocarcinomas, and 9 were squamous cell carcinomas. The tumors of 4 cases had neuroendocrine differentiations, including 2 cases of atypical carcinoid, one case of large cell neuroendocrine carcinoma (LCNEC), and one case of combined LCNEC. A total of 3 tumors were classified as special subtypes of NSCLC including one case of sarcomatoid carcinoma, one case of adenosquamous carcinoma, and one case of combined large cell carcinoma (LCC). A total of 2 further cases showed metastases of extrapulmonary carcinomas in the final histology (1× colorectal adenocarcinoma and 1× squamous cell carcinoma of the uterine cervix). Non-neoplastic or reactive lesions in 11/59 (19%) specimens comprised norm variants in 2 cases (1× intrapulmonal lymph node, 1× bronchogenic cyst) and inflammatory lesions in 7 cases (organizing pneumonia in 4 cases, 1 patient presenting tuberculosis with epitheloid cell granuloma and one case with IgG4-associated inflammatory pseudotumor). Two patients presented unclassifiable necroses in their specimens, and 1 case showed resorptive inflammation in the area of a completely regressive adenocarcinoma without vital residual tumor after neoadjuvant therapy.

### 3.2. Evaluation of Malignancy Based on FCM Scans

A total of 47/48 (98%) neoplastic lesions were correctly recognized as tumors in the FCM scans. The 2 cases of pulmonary hamartoma were correctly grouped as benign tumors, and the solitary fibrous tumor was correctly classified as a spindle cell tumor of uncertain behavior. A total of 44/45 (98%) malignant tumors were diagnosed as carcinomas (sensitivity 98%, specificity 93%, positive predictive value 93%, negative predictive value 93%) with almost perfect levels of agreement with the findings in slides of the FFPE-processed materials (κ = 0.95) and the final histology (κ = 0.91). The case of IgG4-associated pseudotumor was falsely grouped malignant in the FCM scans as well as in the frozen sections due to marked nuclear atypia of stimulated pneumocytes—regardless of the surrounding inflammatory changes. Histology of the FFPE-processed material raised doubts about the diagnosis of adenocarcinoma and the final diagnosis of a reactive process was confirmed in a reference center. Furthermore, subtle infiltrates of mucinous adenocarcinoma were missed in the FCM scans of one case due to a sampling error and the surrounding inflammatory changes leading to a false-negative diagnosis of organizing pneumonia.

FCM was faster than the conventional frozen section technique in all cases. The examinations were carried out by a single person (UT) and always began after the materials for frozen sections had been handed over to the technical assistants. The FCM scans were usually available before the cryostat sections.

The re-evaluation (BT) showed almost perfect agreement with the intraoperative findings (κ = 0.9). The cases with mucinous tumors were evaluated differently by the investigators. Furthermore, the solitary fibrous tumor was classified differently as a reactive fibrosing process, as the macroscopic aspect was not available to the rater.

### 3.3. Histological Subtyping of Non-Small Cell Lung Cancers in FCM Scans

A total of 21/27 (78%) adenocarcinomas (20× primary tumors and 1× metastasis of colorectal cancer) were recognized in the FCM-Scans whereas 6 cases showed solid growth patterns only in the native materials and were provisionally grouped as NSCLC. Furthermore, 8/10 (80%) squamous cell carcinomas (7× primary tumors and 1× metastasis of the uterine cervix) were correctly classified in the FCM. Two cases of primary SCC were grouped as NSCLC when characteristic features were missing in the scans. The recognition of neuroendocrine differentiation was challenging in the native materials. With ideal image quality, one case of atypical carcinoid and one case of LCNEC were correctly classified in the FCM based on their typical nuclear features. In the case of combined LCNEC, only the adenocarcinoma component was named in the scans. Another atypical carcinoid showed regressions in the native material, so this case was classified as NSCLC. The specific subtypes of NSCLC could only be conclusively classified in the histological processing of the FFPE-processed material with the aid of special stains and immunohistology. In the case of adenosquamous carcinoma, only the glandular component was visualized in the FCM scans. The cases of sarcomatoid carcinoma and LCC showed no characteristic features in the FCM scans and were provisionally classified as NSCLC. Overall, 33/44 (75%) of the tumors identified as malignant were classified in moderate agreement with the final histology (κ = 0.59) in the FCM scans. Examinations of frozen sections showed slightly poorer levels of agreement with the final histology (32/45 (71%) cases, κ = 0.55). A higher proportion of SCC and AC were provisionally classified as NSCLC, which may have been due to bias effects in the intraoperative setting.

### 3.4. Morphology of Lung Tissues and Tumors in FCM Scans

Precise imaging of the local normal structures in the FCM was the basis for identifying tumors and premalignant lesions (Figure 2). Bronchial epithelia were easily recognized by their typical columnar shape with basal cell nuclei. The cilia were visualized as a uniform apical rim in reflection mode. The alveolar parenchyma was not uniformly unfolded in the FCM scans, as is the case in sections of the FFPE-processed material. Collagen fibers were more clearly visible in the alveolar septa than in conventional histology. The luminal lining showed flattened type I pneumocytes and cuboidal type II pneumocytes. Macrophages with typical morphology were easily recognizable in the lumina. FCM scans provided a remarkably good visualization of nuclear morphology, which favored the detection of neoplastic proliferation of atypical pneumocytes. The complete replacement of the preexisting alveolar epithelium by a population of enlarged cells was a useful criterion for the diagnosis of lepidic growth patterns of adenocarcinomas. The cytologic malignancy criteria (increased nuclear-cytoplasmic ratio, atypia of nuclear chromatin and nuclear membranes, prominent nucleoli) were clearly visible in the FCM scans. Thus, the detection of atypical pneumocytes was well reproduced in blinded tests.

The histological growth patterns at the tissue level are decisive for the subtyping of lung tumors (Figure 3). Characteristic lepidic and acinar growth patterns of adenocarcinomas were easily recognizable in the FCM scans when present. The prominent visualization of collagen was helpful for the detection of papillary patterns. Particular difficulties arose in FCM examinations of mucinous adenocarcinomas. Due to the low thickness of the optical scans, the method tended to sample errors in these sometimes hypocellular tumors. In such cases, the detection of mucin formation in the scans and in the macroscopic images should lead to the acquisition of several images at different depth levels in analogy to the preparation of deeper sections. Solid variants could only be classified as NSCLC in the FCM if no mucin formation of the tumor cells was visible. For these tumor types, conventional histology with the possibility of PAS staining is superior for subtyping.

The diagnostic criteria for squamous cell carcinoma in conventional (FFPE) histology include intercellular bridges, islands of keratinization, and characteristic architecture. In the FCM scans of the native material, intercellular bridges were not expected as a diagnostic criterion as they are in fact a useful shrinkage artifact of the FFPE procedure. Islets of keratinization were well recognizable in the scans as round structures with a strong (red) signal in reflection mode. Furthermore, the architecture in band-like patterns with the characteristic mosaic-like arrangement of trapezoidal cells in combination with clearly recognizable nuclear atypia was a valid pattern for the detection of squamous cell carcinoma in FCM. In basaloid PEC, in which these characteristic patterns were not developed, squamous differentiation could only be demonstrated in the FFPE-processed material by immunohistochemistry.

Neuroendocrine differentiation is characterized by the typical pepper-and-salt structure of the nuclear chromatin, which was also clearly visible in the FCM scans of the well-differentiated tumors. The subtle histological growth patterns with the formation of pseudo-rosettes were only recognizable in one case. In the other cases with neuroendocrine tumor differentiation in the present study, there was morphological overlap with basaloid squamous cell carcinoma or solid NSCLC, so the diagnosis of atypical neuroendocrine tumor or large cell neuroendocrine carcinoma could only be made together with immunohistology.

Mesenchymal lesions were easily distinguished from lung carcinomas in the FCM scans (Figure 4). Pulmonary hamartomas demonstrated a characteristic structure already in the macro images and were easily recognizable due to their composition of mature cartilage, differentiated adipose tissue, and enclosed atypia-free bronchial epithelia. The solitary fibrous tumor was recognizable in the FCM images as a spindle cell tumor with a characteristic growth pattern, typical features of the tumor cells, and the collagen fiber-containing matrix. Immunohistological examinations were performed to differentiate this tumor from other mesenchymal tumor entities.

Some of the unclear tumor-suspect findings were correctly classified as reactive lesions in the FCM (Figure 5). The fibroblast foci in organizing pneumonia were reliably recognizable in the FCM on the basis of their fibrous stroma and the inconspicuous spindle cells. The inflammatory infiltrates as an important criterion of inflammatory lesions could be reliably assessed in the FCM scans in terms of their extent and distribution. The cytologic criteria of granulocytes, lymphocytes, macrophages, and giant cells were consistent with the final histology. Necrosis zones were visualized as homogeneous granular areas with uniform intensity in the reflectance and fluorescence channels. Thus, epithelioid cell granulomas in a patient with tuberculosis could already be reliably diagnosed in the FCM.

### 3.5. Comparison of FCM, Frozen Sections and Conventional Histology

The diagnoses of FCM, FS, and HE-stained paraffine sections are summarized in Table 2. There were minor differences in the determination of biologic behavior. All three methods showed almost perfect levels of agreement with the final diagnosis (κ = 0.91 for FCM, κ = 0.95 for FS, and κ = 1.0 for paraffine slides). A total of 44/45 (98%) carcinomas were recognized as malignant in all three approaches and 13/14 (93%) lesions were consistently classified as benign (sensitivities 98–100%, specificities 93–100%). Differences were found in 2 cases. The IgG4-associated pseudotumor was consistently classified as malignant in FCM and FS. The prominent presentation of (reactive) nuclear atypia led to false positive diagnoses in the intraday examinations. The nuclear changes were less pronounced in the HE sections after the FFPE procedure. In retrospect, the prominent inflammatory infiltrates should have called for caution in the diagnosis of a malignant tumor. The sampling error of the FCM in the case of a mucinous adenocarcinoma was avoided in FS and paraffin sections due to greater slice thicknesses (5 vs. 0.5 µm in the FCM) and a greater penetration depth.

All three methods also showed comparable results for the histological subtyping of the tumors. Overall, there was moderate agreement between the FCM, FS, and the slides of the paraffin-embedded materials with the final tumor types. In 25–33% of the tumors, representative sampling and additional immunohistological techniques were necessary for final typing. In our experiments, squamous cell carcinomas were more frequently correctly classified in the FCM and FS than in the examination of FFPE-processed material. It can be concluded from this that the (artificial) formation of intercellular bridges was a less reliable criterion for typing and that there were morphological overlaps with other tumor types in the paraffin sections. Adenocarcinomas were predominantly correctly classified on the basis of FCM scans, FS, and conventional H&E-slides (75% in FCM and FS and 82% in paraffin sections). Neuroendocrine differentiation was less reliably recognized in FCM and FS than after the FFPE procedure. The result is not very meaningful due to the small number of cases. In the classic case, the typical nuclear changes were very well represented in the FCM scans. The error rates for the special types are given by technical constraints. These are usually combined tumors or rare entities with pronounced heterogeneity that require extensive sampling and were systematically not represented in the intraoperative setting when examining one or a few samples.

### 3.6. Summary

Lung tumors could be diagnosed intraoperatively with FCM with the same certainty as with frozen sections. FCM examinations required only minimal technical effort due to pretreatment with the fluorescent dye. In contrast to the labor-intensive preparation of frozen sections, the digital scans could be recorded by one person without much effort. After FCM, the examined tissue was available without loss as unfixed material for further analyses.

## 4. Discussion

With the introduction of a screening program for lung tumors, it is expected that carcinomas will be detected at an earlier stage. The further increase in the importance of surgical procedures for the treatment of localized tumors will pose particular challenges for the intraoperative classification of lung tumors. In particular, the differentiation of preinvasive from early invasive lesions in frozen sections is currently a field of intensive work in pathology [33] with the aim of implementing intraoperative immunohistological techniques [34].

Ex vivo FCM is a promising alternative to frozen sections for the intraoperative diagnosis of lung tumors. Our results documented the excellent suitability of FCM scans for evaluations of malignancy with a performance comparable to that of intraoperative frozen sections. The investigators noticed a remarkably good presentation of the nuclear features in the FCM scans of the native material outperforming frozen sections by far. In particular, the cytological malignancy criteria (nuclear-cytoplasmic ratio, structure of the nuclear chromatin, nucleoli) are clearly recognizable on the digital scans. If present, the diagnostically relevant criteria of the individual tumor entities (horn beads in squamous cell carcinomas, glandular architecture in adenocarcinomas) are also clearly recognizable.

In the retrospective analysis, ex vivo FCM showed comparable results to frozen sections with a comparable agreement of the intraoperative diagnoses with the final examination of the FFPE-processed materials. However, the direct comparison of FCM scans and frozen sections of the same material was not previously included in the study design in order to avoid delays in routine diagnostics and should be documented in follow-up studies. The acquisition of the FCM scans was less time-consuming in terms of personnel and technology and could be carried out by a single person in a comparable time. The histological representation of the tissue in the FCM scans was less susceptible to technical limitations such as tissue shrinkage, tear artifacts, and ice crystals. However, uneven cut surfaces of the examined samples easily lead to missing areas in the scans, as the thickness of the optical sections is just 0.5 µm and a maximum penetration depth of 200 µm can be achieved. In mucinous tumors, only a few representative tumor cells were visualized in the FCM, which led to a sampling error in one case. Conventional frozen sections had an advantage here, as a larger tumor volume could be sampled by additional serial sections.

The main limitation of our study is the small number of cases. Further studies on larger numbers of patients and involving as many centers as possible are needed before the technique can be established in the intraoperative routine diagnosis of lung tumors. However, the promising results of this first feasibility study focusing solely on lung resectation specimens suggest that the previous experience of our working group with FCM on samples from the prostate [21,25] and the liver [26] might be transferred to the diagnosis of lung tumors. We hope that this study will encourage other lung cancer centers to gain diagnostic experience with this instrument, which has advantages over frozen sections due to its material-sparing properties.

Frozen sections are associated with significant tissue loss for technical reasons, which is a serious problem when examining smaller lesions. In contrast, the tissue is available as largely untreated native material without any loss after FCM analysis, which is an advantage over frozen sections. Ex vivo FCM thus represents a promising instrument for the collection of biobank samples. Based on the high-resolution scans, representative tumors and normal tissue can be obtained in a targeted manner, even from small, macroscopically barely visible lesions. Furthermore, loss-free imaging enables effective quality control prior to permanent preservation of the tissue. According to the current state of knowledge, pretreatment with AO is not expected to have any disadvantages for subsequent molecular examinations, as the dye is used for live examinations of cell cultures [35]. However, the substance is considered to be potentially mutagenic, and further sequencing analyses are necessary to ensure its suitability in molecular pathology.

Our results suggested a potential role of the method in the pre-therapeutic biopsies for histological confirmation of lung cancer, e.g., in EBUS samples. The excellent visualization of nuclear malignancy criteria underlines the particular suitability of ex vivo FCM also for examinations of cytological materials [36], which has already been used in recent studies on needle aspirations from the pancreas and thyroid gland [37,38]. According to verbal communication with the developers, algorithms for immunofluorescence techniques are also currently being implemented. Using antibodies suitable for fluorescence, a better classification of lung tumors could be achieved in the intraoperative setting.

The VivaScope 2500-G4 used in the study made it possible to output the pseudo-colorized scans as digital image data in cross-platform formats. The method thus formed a suitable basis for telepathological second opinions for external specialists. The image data were very well suited for automated digital image analysis with common, freely available analysis programs and represent an interesting basis for automated analysis procedures [39,40,41].

## 5. Conclusions

Ex vivo FCM proved to be a potential alternative to frozen sections in the intraoperative diagnosis of lung tumors. The results in the assessment of biologic behavior and subtyping are comparable. FCM is a material-sparing procedure that can be carried out quickly and easily by one person. The material is preserved without loss as native material for further examinations and for biobanking. As digital data, the images are an optimal basis for telepathological applications, for example, external second opinions.

## Figures and Tables

**Figure 1 cancers-16-02221-f001:**
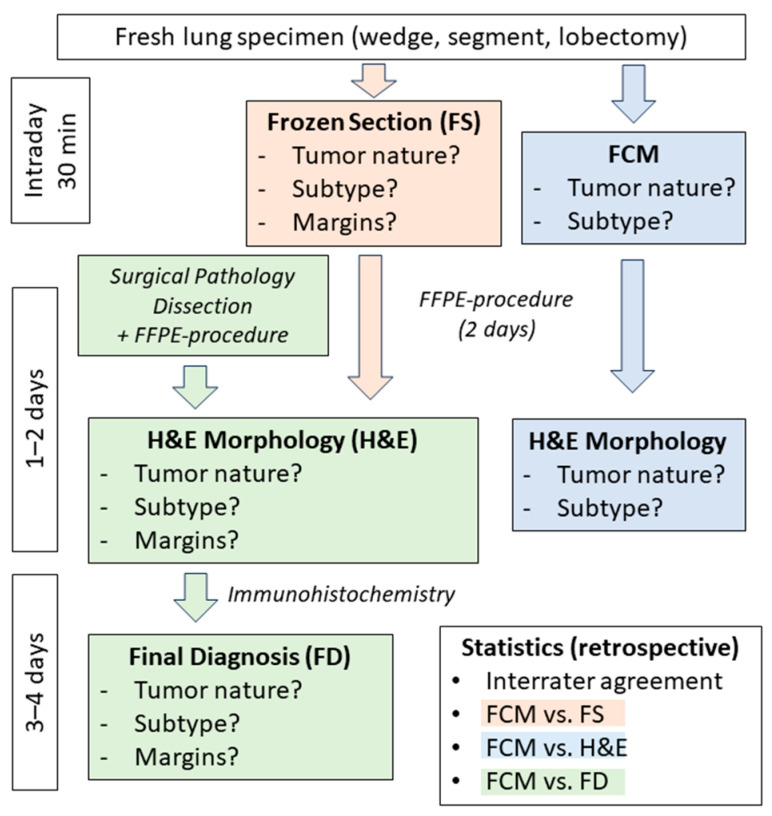
Study design.

**Figure 2 cancers-16-02221-f002:**
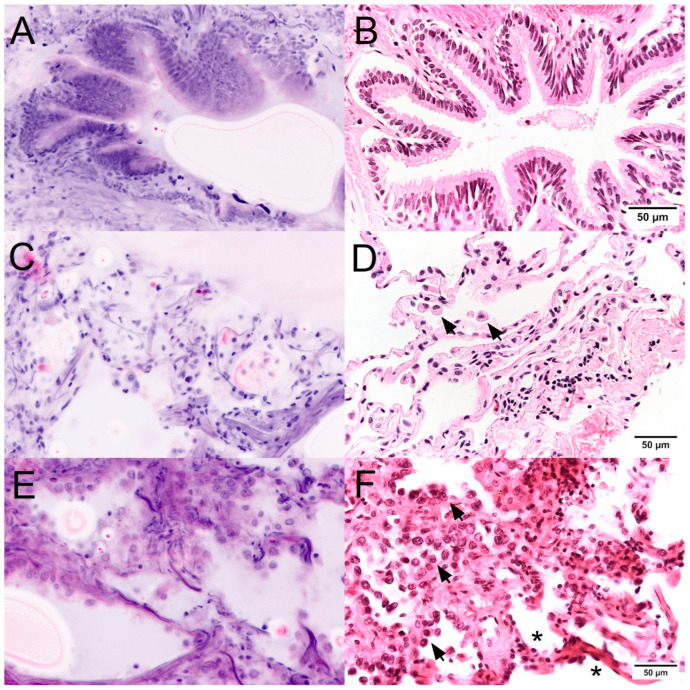
Malignancy in FCM scans (**left**) and conventional histology (**right**). Bronchial epithelia (**A**,**B**) were easily recognizable. The parenchyma (**C**,**D**) showed flattened or cuboidal pneumocytes and luminal macrophages (arrows). The clear presentation of nuclear features (**E**,**F**) favored the detection of neoplastic proliferations of atypical tumor cells (arrows) in comparison to nonneoplastic pneumocytes (*).

**Figure 3 cancers-16-02221-f003:**
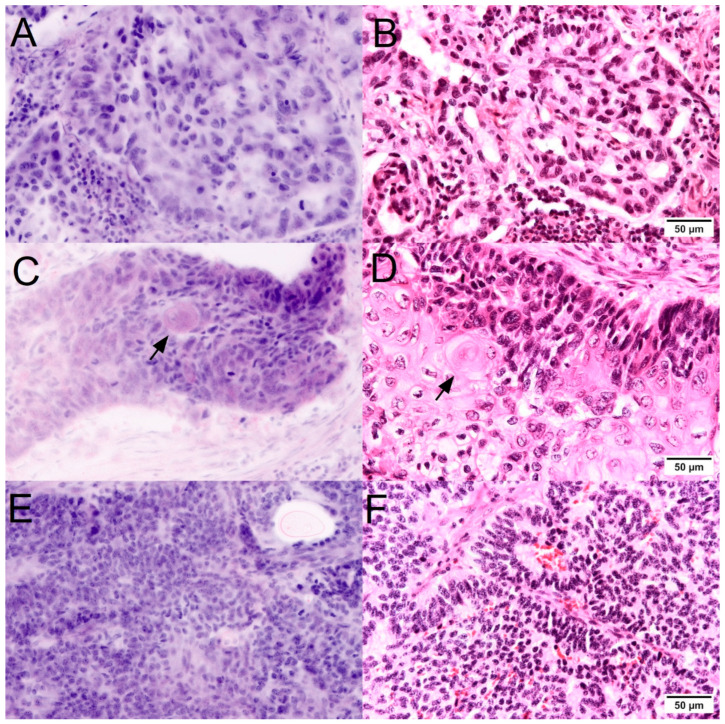
Lung tumors in FCM scans (**left**) and conventional histology (**right**). Characteristic lepidic and acinar growth patterns of adenocarcinomas (**A**,**B**) were easily recognizable in the FCM scans. Islets of keratinization (arrow) were well recognizable in squamous cell carcinomas (**C**,**D**) as round structures with a strong (red) signal in reflection mode. When present, the typical pepper-and-salt chromatin structure and so-called pseudo-rosettes were recognizable hallmarks of neuroendocrine tumors (**E**,**F**).

**Figure 4 cancers-16-02221-f004:**
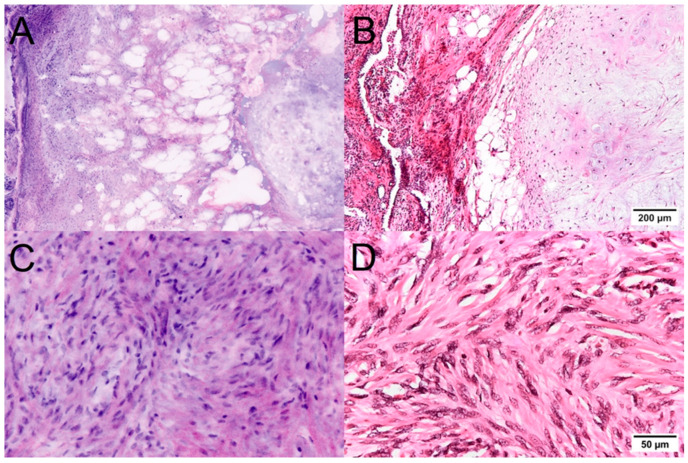
Mesenchymal lung tumors in FCM scans (**left**) and conventional histology (**right**). Pulmonary hamartomas (**A**,**B**) were easily recognizable by their composition of mature cartilage, differentiated adipose tissue, and enclosed atypia-free bronchial epithelia. In the case of a Solitary fibrous tumor (**C**,**D**), the spindle cell morphology and the mild degree of atypia were recognizable in the FCM scans. Interestingly, the fibrous matrix was also clearly visible.

**Figure 5 cancers-16-02221-f005:**
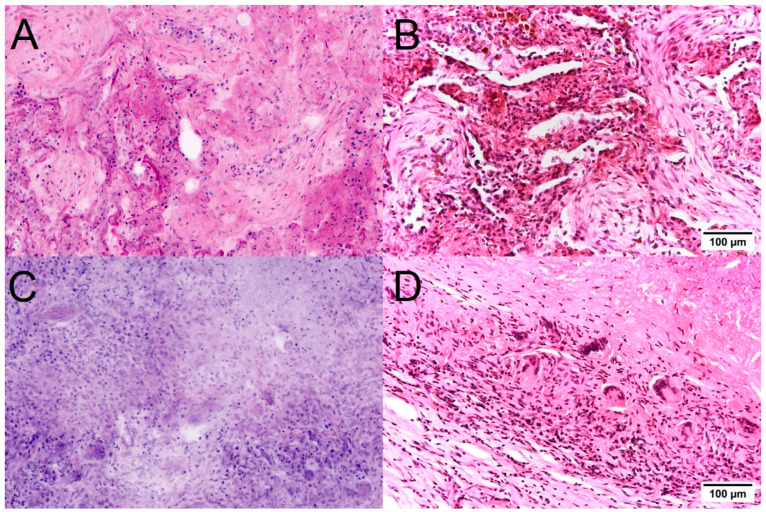
Inflammatory lesions in FCM scans (**left**) and conventional histology (**right**). Fibroblast foci in organizing pneumonia (**A**,**B**) were reliably recognizable in the FCM on the basis of their fibrous stroma and the inconspicuous spindle cells. The cellular composition and the zonal organization of an epithelioid cell granuloma (**C**,**D**) were clearly shown.

**Table 1 cancers-16-02221-t001:** Overview of the enrolled cases.

Reactive/Inflammatory Lesions		11
Intrapulmonary Lymph node	1	
Bronchogenic Cyst	1	
Organizing Pneumonia	4	
Epitheloid cell granuloma	1	
IgG4-associated inflammatory Pseudotumor	1	
Residual Inflammation after Therapy (ypT0)	1	
Necrosis, NOS	2	
**Benign neoplastic lung tumors**		3
Pulmonary Hamartoma	2	
Solitary fibrous tumor	1	
**Malignant neoplastic lung tumors**		43
Squamous cell carcinoma	9	
Adenocarcinoma	27	
Neuroendocrine Tumors	4	
Atypical Carcinoid (n = 2)		
Large Cell Neuroendocrine Carcinoma (n = 1)		
Combined LCNEC (n = 1)		
Other types of NSCLC	3	
Sarcomatoid Carcinoma (n = 1)		
Combined SCC + LCC (n = 1)		
Adenosquamous carcinoma (n = 1)		
**Metastases**		2
Colorectal (Adenocarcinoma)	1	
Uterine cervix (Squamous cell carcinoma)	1	

**Table 2 cancers-16-02221-t002:** Diagnoses from FCM, frozen sections, and conventional histology.

	FCM		Frozen Sections		H&E		FD
	Benign	Malignant	Benign	Malignant	Benign	Malignant	
Benign	13	1	13	1	14	0	14
Malignant	1	44	0	45	0	45	45
Sensitivity	98%		100%		100%		
Specificity	93%		93%		100%		
Positive p.v.	98%		98%		100%		
Negative p.v.	93%		100%		100%		
Cohens Κ	0.91		0.95		1.00		
SCC							
correct	8		6		3		10
“NSCLC”	2		4		6		
false	0		0		1		
missed	0		0		0		
Adeno-Ca							
correct	21		21		23		28
“NSCLC”	6		7		4		
false	0		0		1		
missed	1		0		0		
Neuroend.							
correct	2		2		3		4
“NSCLC”	1		2		1		
false	1		0		0		
missed	0		0		0		
Spec. types							
correct	2		3		1		3
“NSCLC”	0		0		0		
false	1		0		2		
missed	0		0		0		
Cohens Κ	0.59		0.56		0.45		

## Data Availability

The data are not publicly available due to privacy restrictions. The data presented in this study are available upon reasonable request from the corresponding author.

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
