# Peer review of "Ex Vivo Fluorescence Confocal Microscopy for Intraoperative Examinations of Lung Tumors as Alternative to Frozen Sections—A Proof-of-Concept Study"

_cancers, 2024, doi:10.3390/cancers16122221_

Round 1

Reviewer 1 Report

Comments and Suggestions for Authors

The aim of this work is to test the suitability of Ex Vivo fluorescence confocal microscopy for use for intraoperative diagnosis of lung tumors. The result suggests that FCM provide almost similar results to frozen section in predicting the presents of cancer in a given sample. If implemented this method can  revolutions lung surgeries as it will remove the need for frozen section. Overall the writers make a good clam for the suitability of this method to replace FC in most cases of lung tumor surgery

The two minor noted I have are:

·       I don’t understand the meaning of word "dignity" in the context of H&E staining.

·       AO have several excitation wave lengths, which was used in the VivaScope 2500-G4

Reviewer 2 Report

Comments and Suggestions for Authors

In this study authors performed ex vivo FCM and demonstrated a potential alternative to frozen sections (FS) in the intraoperative diagnosis using tissues from the lung tumors. Ex vivo tissue imaging studies were compared with FCM, IHC and FS. Overall study highlighted FCM data results were comparable (Table 1) to FS, while FCM can be carried out quickly and easily. In addition, specimen material can be preserved without loss from native form. However, the key weakness of this study is limited number of samples size, and it was tested in only one tumor model. Although FCM could be an efficient technique, but it requires more rigorous testing approach. Hence, one cannot conclude FCM can be used in routine clinical diagnosis. I would recommend, this manuscript can be accepted after inclusion of additional data sets performed with one more tumor model as like current studies, e.g., GBM or Breast cancer.

Comments on the Quality of English Language

Good
